# Assessing the ecological impacts of transportation infrastructure development: A reconnaissance study of the Standard Gauge Railway in Kenya

**Tobias Ochieng Nyumba**[1,2]*, **Catherine Chebet Sang**[2], **Daniel Ochieng Olago**[2], **Robert Marchant**[3], **Lucy Waruingi**[1], **Yvonne Githiora**[2], **Francis Kago**[1], **Mary Mwangi**[1], **George Owira**[2], **Rosemary Barasa**[2], **Sherlyne Omangi**[2]

**1** African Conservation Centre, Nairobi, Kenya, **2** Institute for Climate Change and Adaptation, University of Nairobi, Nairobi, Kenya, **3** Department of Geography and Environment, The York Institute for Tropical Ecosystems, University of York, North Yorkshire, United Kingdom

* tnyumba@uonbi.ac.ke

**Data Availability Statement:** All relevant data are within the manuscript and its Supporting Information files.

## Abstract

Transportation infrastructure, such as railways, roads and power lines, contribute to national and regional economic, social and cultural growth and integration. Kenya, with support from the Chinese government, is currently constructing a standard gauge railway (SGR) to support the country's Vision 2030 development agenda. Although the actual land area affected by the SGR covers only a small proportion along the SGR corridor, a significant proportion of the area supports a wide range of ecologically fragile and important ecosystems in the country, with potential wider impacts. This study used a qualitative content analysis approach to gain an understanding and perceptions of stakeholders on the potential ecological impacts of the interactions between the SGR and the traversed ecological systems in Kenya. Three dominant themes emerged: 1) ecosystem degradation; 2) ecosystem fragmentation; and 3) ecosystem destruction. Ecosystem degradation was the most commonly cited impact at while ecosystem destruction was of the least concern and largely restricted to the physical SGR construction whereas the degradation and fragmentation have a much wider footprint. The construction and operation of the SGR degraded, fragmented and destroyed key ecosystems in the country including water towers, protected areas, community conservancies and wildlife dispersal areas. Therefore, we recommend that project proponents develop sustainable and ecologically sensitive measures to mitigate the key ecosystem impacts.

## Introduction

The contribution of transportation infrastructure to economic, social and cultural growth through regional and global integration has had mixed outcomes [1–3]. There has been unprecedented growth and expansion of transportation infrastructure, particularly in sub-

**Funding:** All the authors of this paper are jointly funded under the Development Corridors Partnership (DCP) project. The DCP is a UK Research and Innovation's Global Challenges Research Fund (GCRF) funded project (project number: ES/P011500/1). The DCP aims to generate decision-relevant evidence and feeding into key decision-making processes in order to improve the sustainable development outcomes and investments in infrastructure projects.

**Competing interests:** The authors have declared that no competing interests exist, financial or otherwise.

Saharan Africa, accounting for up to 90% of new infrastructure including but not limited to railways, roads and power lines [4, 5]. Recently, many African governments have increasingly favoured railways due to their economic and environmental advantages [6]. Since their inception, railway technology has evolved considerably and reduced transportation costs and increased safety not only for passengers, railway workers and freight but also for the human populations living within the railway corridors [7, 8]. Despite the advantages, and the need to meet the increasing demand for environmental accountability, there is a growing recognition of their impacts on the natural environment, especially in remote and fragile ecosystems characterized by low human population, poor and marginalized communities and marginal or changing climatic conditions. Furthermore, although railways may share impacts with other anthropogenic activities, they have unique impacts associated with their linear form constituting "disturbance corridors" that disrupt the natural, more homogeneous landscape [9: 157]. Consequently, new calls are emerging to identify, quantify and mitigate the impacts of railways and other transportation infrastructure [4, 10–15]. The potential ecological impacts of the railways can be captured through stakeholder perceptions, and that insight used to inform on ecologically sensitive design, implementation, and mitigation of linear infrastructure impacts.

The rapidly expanding transportation infrastructure can impact the environment both directly, as an immediate consequence of the presence of the infrastructure and its construction or indirectly, as a result of human activities that are facilitated by new infrastructure [13]. An analysis of the impacts of 33 planned and existing development corridors, including transportation infrastructure in sub-Saharan Africa showed that ecosystems have been significantly impacted by increasing land-use pressure and encroachment that has in part been triggered by the infrastructure development [13]. Other studies have established that transportation infrastructure leads to loss of ecosystem integrity through truncation of ecosystems into smaller, often isolated, patches that may not be able to maintain or sustain ecological processes in the long run [16]. Such impacts include bisection of watersheds and basins; physical disturbance and disruption to the composition, structure and functioning of ecosystems, and movement, migration and survival of resident wildlife species [13, 14, 17]; direct mortality of wildlife through vehicle/train–wildlife collisions (V/TWC) [10, 16, 18], and behavioural modification among diverse species in different ways [19]. Furthermore, it is argued that transportation infrastructure contributes to soil, water and air pollution [20]; alteration of natural processes including natural hydrology, fire regimes and competitor and predator-prey relationships among other impacts [21–25]. According to Hulme [26] and Catford *et al.* [27], transport infrastructure may act as migration corridors for the natural dispersal of non-native biodiversity by allowing their movement across physical and environmental barriers or by supplying suitable habitat for their expansion.

The most severe but rarely reported impact of transportation infrastructure is the destruction and loss of natural ecosystems [28]. Ecosystem destruction is the absolute loss of habitat surface area through the physical presence of a road or railway and related facilities that can take up a substantial amount of space [20]. The construction of transportation infrastructure often results in land use conversion, from natural ecosystems such as forests and water bodies, to transportation land use or right-of-way [21, 29]. This entails clearing of vegetation and the accompanying levelling operations that destroy the original topography and soil profile. It further entails the elimination, replacement or modification of the original characteristics of natural vegetation and aquatic ecosystems [20, 21].

Previous research on the ecological impacts of infrastructure has focused largely on roads where a considerable effort has been expended to quantify their effects particularly in Europe, North America and Australia [30]. Although railways share similar ecological impacts with other transportation infrastructure, little research has been undertaken. Hence, less is known

about the direct and indirect impact of railways on ecological communities and processes. Understanding the impacts of railways and the associated rail traffic is important as part of Kenya's commitment to international initiatives for the protection of biological diversity, supporting the evaluation of the effectiveness of impact mitigation measures, and to support cumulative environmental assessment and transportation planning.

To characterise and assess these impacts, qualitative content analysis was used to capture the rich and deep narratives from a wide range of stakeholders [31]. Qualitative data was collected from 19 group interviews and meetings comprising 54 key informants from 14 sites along the SGR phases 1 and 2A stretching from the Kenyan coastal city of Mombasa to Suswa in Rift Valley. To ensure we captured the wide range of impacts and benefits we emphasized trust, transparency, verifiability and flexibility in our method [32, 33]. We used Qualitative Data Analysis Miner Lite (QDA) software to code and categorize the data. ArcGIS 10.4 was used to spatially map the SGR, key ecosystems and protected areas; the GIS was queried together with the qualitative data to identify impact hotspots.

## Approach and methodology

This study was approved by the Ethics Review Group of the UNEP-WCMC and the ESRC following the Code of Practice on Ethical Standards in Research. The protocols used in the study were approved by the National Council for Science and Technology and Innovation of the republic of Kenya (Permit No. NACOSTI/P/19/3232/27585). A total of 54 state and non-state officials were interviewed between January and May 2019 along the entire SGR Corridor in southern Kenya. Informed consent was sought according to the UNEP-WCMC and ESRC Research Ethics guidelines and strategies aimed at minimizing harm to the subject.

### Study area

The study was conducted along the entire stretch of the Kenya Standard Gauge Railway (SGR) Phase I & Phase IIA, covering eight counties from Mombasa to Narok (Fig 1). The SGR has been billed as the biggest transport infrastructure in the country's history under the Vision 2030 development agenda [34]. The SGR forms part of the East African Railway Master Plan (2009) and the Eastern African SGR regional network which aims to rejuvenate existing railways serving Tanzania, Kenya, Uganda and extend to Burundi, Rwanda, Ethiopia and South Sudan [35]. The SGR runs westwards from the coastal town of Mombasa and through the central Kenya with the line through western Kenya to Malaba town at the Kenya-Uganda border still under construction to link up with other standard gauge railways that are being built in East Africa [35]. The construction of the SGR begun in 2014 at an estimated cost of US$3.8 billion, with 90% supplied by a loan from the Exim Bank of China and 10% coming from the Kenyan government [34].

The prime contractor on the railway was the China Road and Bridge Corporation [5]. The construction of the SGR has been undertaken in phases: the first phase from Mombasa to Nairobi was completed and has been in operation since May 2017, while the second phase from Nairobi South Railway Station to Naivasha Industrial Park in Enoosupukia and onto Narok town was completed in August 2019. Meanwhile, the third phase covering Narok to Kisumu and onto Malaba is yet to be constructed [5].

The SGR is categorized as a National Class I railway and has a wide range of safety protection measures in the design and operation that include speed limits, installation of high guard fence, safety buffers and earth embankments to avoid crossing other infrastructures. Furthermore, bridges, underpasses, culverts and flyovers have been constructed in wildlife areas such as Tsavo and Nairobi National Parks and in high human density areas such as Athi River to

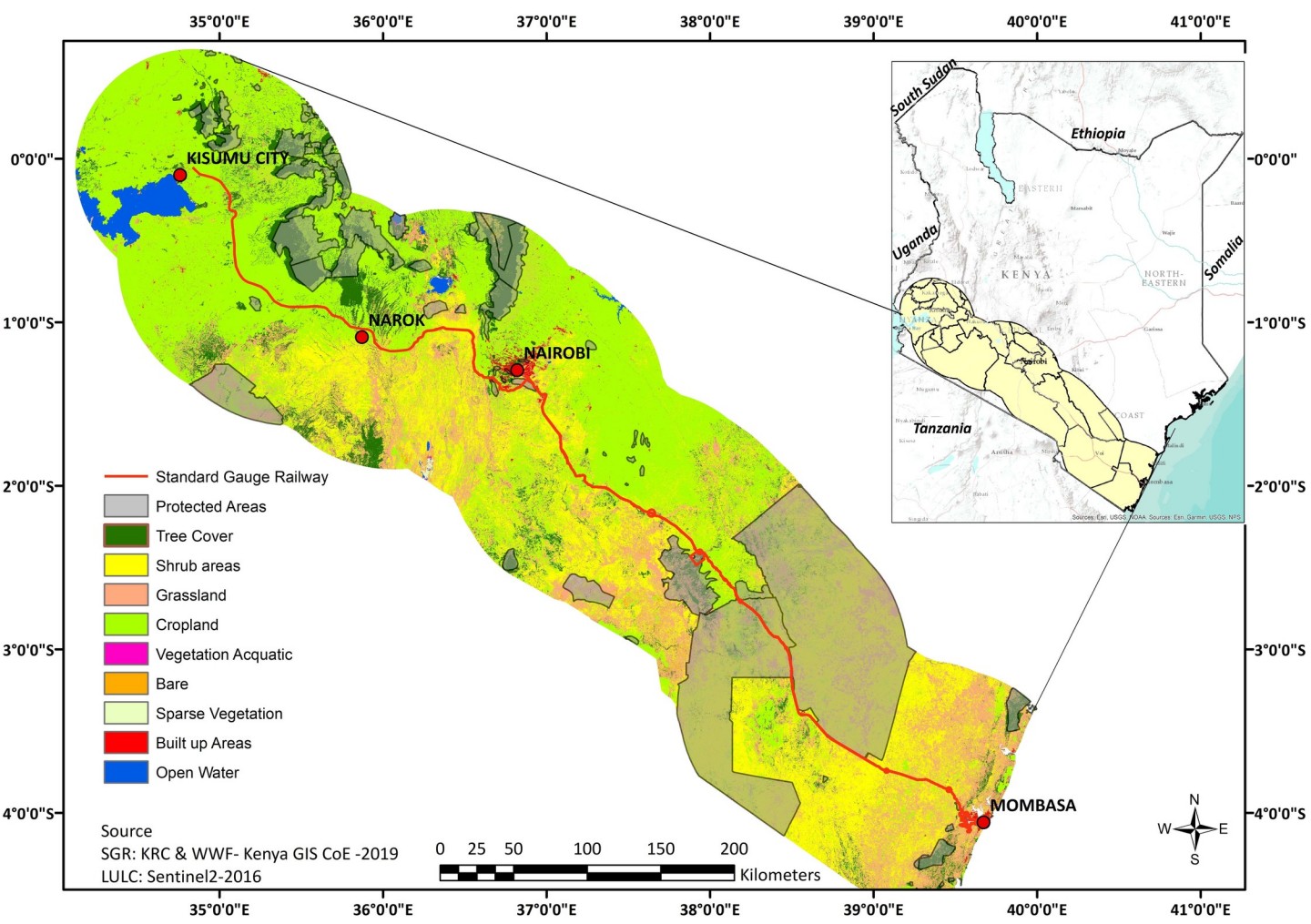

**Fig 1. Map of SGR corridor and different resources.**

facilitate free movement of wildlife and people. Within Nairobi National Park, an acoustic noise barrier has been installed to reduce noise disturbance to wildlife [36, 37]. Despite the measures taken to minimise the impacts of the SGR, anecdotal reports point to the existence of negative impacts on the ecological communities and processes along the SGR corridor.

## Data collection

The study gathered insights and perspectives from group interviews and meetings with a diverse range of stakeholders. Individual interviews were also carried out with experts who were either working alone or had colleagues out in the field. Interview questions revolved around the description of their mandates and interaction with the SGR and participants' perceptions. The participants were drawn from the corridor institutions such as Kenya Railways Corporation and Kenya Ports Authority (n = 20), government institutions mandated with natural resource conservation and management such as the National Environment Management Authority, Kenya Forest Research Institute, Water Resources Authority and the Kenya Wildlife Service (n = 10), community groups such as Community Forest Associations, local farmers

and pastoralists (n = 10), non-governmental and research organisations (n = 4) and three county governments along the SGR (n = 10).

All the interviews were recorded and reviewed at the end of the day. In addition, all the activities within and around the SGR installations, the status of surrounding landscape, resources and participants' conversations were observed, captured and recorded for further analysis. The observations focused on visible and verifiable conditions of the landscapes that could be or were linked to the SGR activities such as construction and maintenance as well as the relationship and behaviour of stakeholders while discussing cross-cutting issues.

## Data analysis

Qualitative content analysis was used to identify the potential ecological impacts of the SGR based on the perceptions of stakeholders. This interpretive process focuses on both the subject and background and explores the similarities and differences between and within different parts of the text [38]. Computer-Assisted Qualitative Data Analysis software (CAQDAS) called Qualitative Data Analysis (QDA) Miner® [39] was applied to code and categorize the qualitative data into thematic areas, and preliminary codes that were based on the literature [40] and appropriate categories identified through the analysis of the field data. Codes and units of meaning were interpreted in the context of the study and compared for similarities and differences (see S1 and S2 Appendices). All the geo-referenced data were transferred to the Geographic Information System (GIS) for spatial analysis using ArcMap in ArcGIS 10.4 version [41].

## Results on the ecosystem impacts of the SGR

The results of our analysis and inspection of codes and subclasses resulted in the extraction of three main themes: ecosystems degradation, ecosystem fragmentation and ecosystem destruction.

### Ecosystem degradation

Ecosystem degradation emerged as the main category of impact during the meetings and consisted of three subcategories.

**Contamination of soil, water and air.** Participants in most of the meetings identified issues around soil, water and air contamination during construction and operation of the SGR. Expected impacts of the SGR included population growth through migration, shifts in land use and land tenure and the emergence of illegal activities such as illegal grazing in protected areas. Officials of the Kenya Wildlife Service observed that "*local communities [were] using the underpasses to pass their livestock through to Tsavo National Park particularly around Buchuma gate*". The livestock incursions they observed "*resulted in serious soil degradation in the southern part of Tsavo East*". Other concerns included oil spills as observed by local officials in Kibwezi County "*pollution of the Thange river [though oil spill of 2015 and a recent one in May 2019 around Machakos that contaminated Athi River] has had a great impact on the community given that it provides water for cultivation which is the main economic activity in the area. Since the oil spill incident, the use of the [Thange] river for irrigation, livestock watering, and domestic purposes have been suspended and the land in the affected area is still unsafe for cultivation*".

Participants also observed that noise pollution during construction and operation of the SGR were common in the areas around Nairobi and Voi. Officials in Voi stated that "*the main environmental impact currently observed [by the KRC staff] is noise pollution when the trains are passing*, while local communities around Nairobi county reported impacts of "*blasting for*

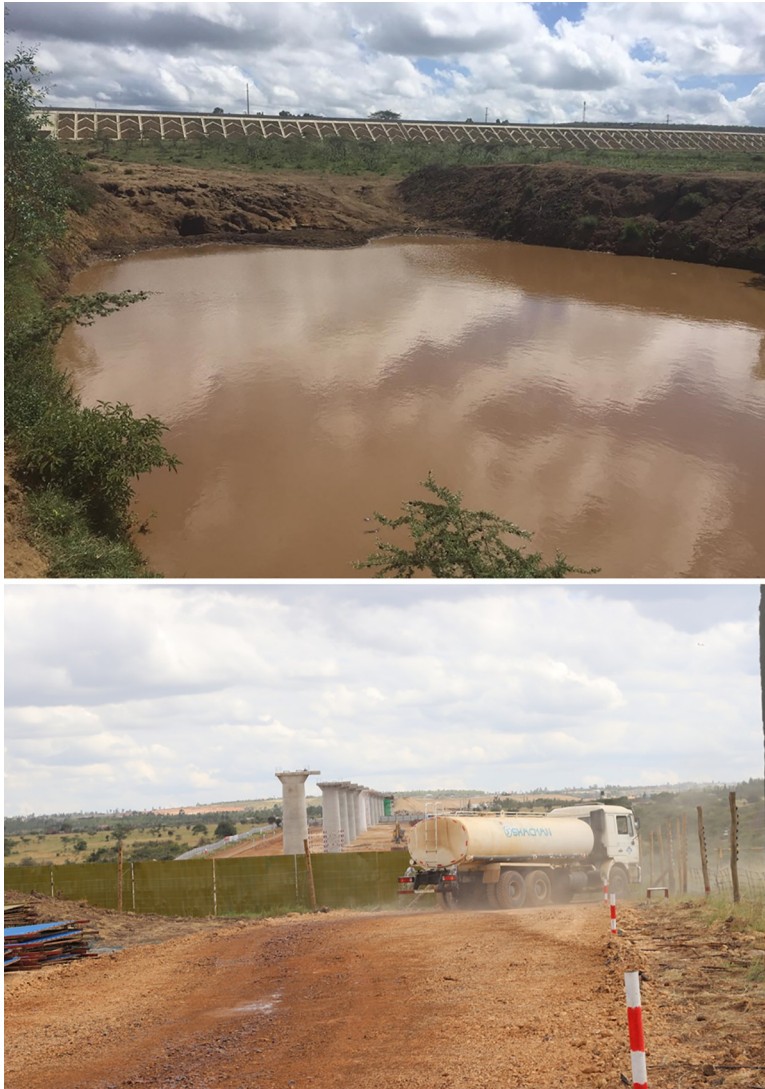

**Fig 2. A new water body with high suspended sediment load in an abandoned quarry (a) and dust pollution during construction in Tuala area (b).**

construction materials causing tremors in the area and leading to buildings cracking, for example, at Oloosirkon primary school. Meanwhile, dust pollution [was] also a challenge and impacts include infections from dust, coughs and chest pain" (Fig 2).

**Soil erosion, sedimentation and flooding.** This was mainly reported in areas along the coast where local Community Forest Association officials observed that sediments eroded from the rail embankments *"did not only affect mangroves seed development and self-germination but also blocked streams and reduced the stream size in Kilindini"*. Meanwhile, respondents from Narok county observed that *directing water to the underpasses led to gulley erosion* (Fig 3A) *affecting soil cover and leading to siltation of Lake Magadi*. In Voi, county officials observed that *"storm water directed to the culverts flooded low lying homesteads and farms during heavy rains"*. Furthermore, respondents from Nairobi and Narok Counties reported incidents of *"flooding along the culvert [underpasses] when it rained while rivers [Empakashe and Mbagathi] had been blocked or dried up completely because they had been filled with silt from the*

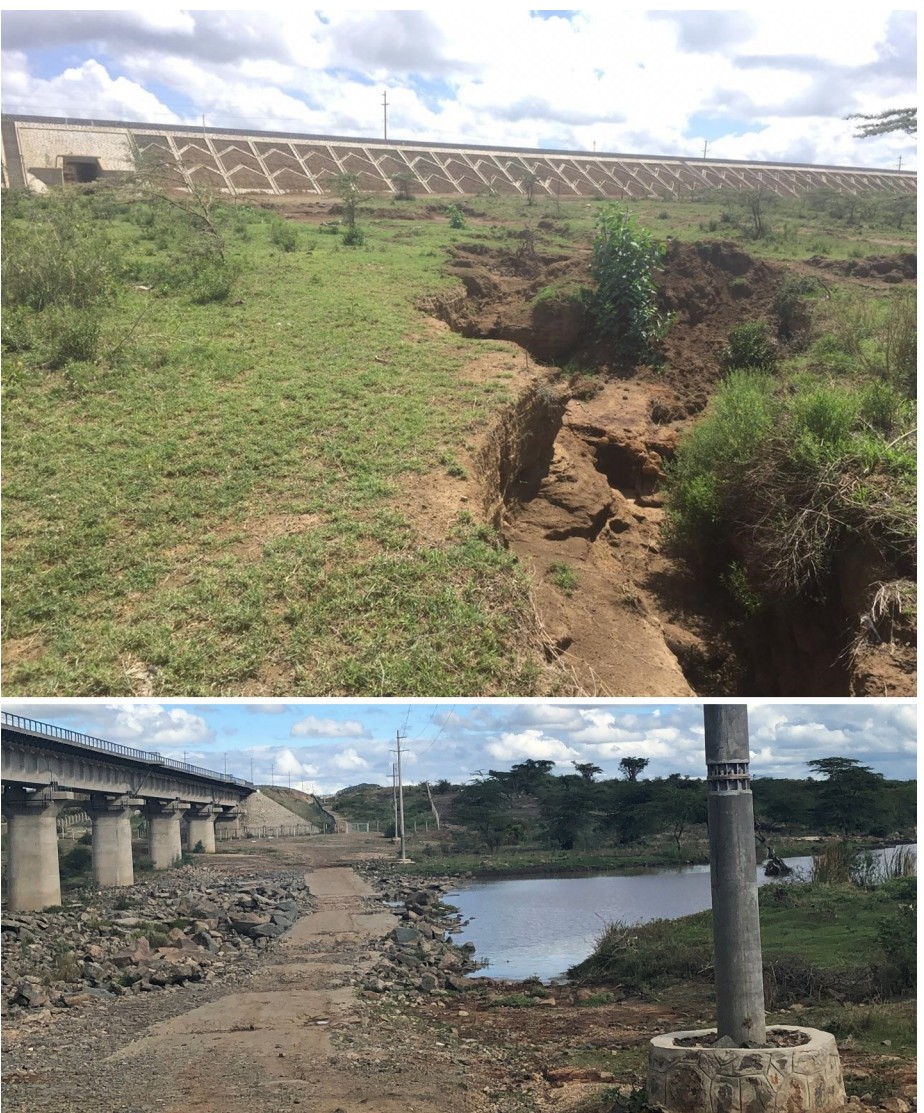

**Fig 3. A damaged stream in Kitengela (a) and an eroded area near an underpass in Duka Moja (b).**

*construction*". Other areas affected by flooding due to redirected water included Kibarani, Kiunduani, Ngwiw'a, Mutantheeu, Emali town and Kima in Makueni County.

**Introduction and spread of invasive plant species.** Participants from Voi reported that invasive plant species had recently emerged and were spreading rapidly along the SGR corridor. This, they observed "*was a problem in both [Tsavo East-West] National Parks where the invasive cactus Opuntia stricta and Prosopis juliflora (known locally as Mathenge) was also prevalent along the new highway from Voi to Taveta*". However, we could not establish the connection between the spread of invasive plant species and the construction and operation of the SGR. Meanwhile, in some areas in Kajiado county, invasive plant species were regenerating after long periods of absence due to what locals observed as "*disturbance through borrow pits and truck tracks taking soil to the SGR construction sites*". Although our study team made clear observations of the presence of these invasive plant species, we could not verify that they resulted from the construction and operation of the SGR. This was because there were no clear

timelines or monitoring on the emergence and spread of these species linking them to the construction activities.

## Ecosystem fragmentation

Ecosystem fragmentation emerged as the second dominant theme from our consultative meetings. The SGR traversed key ecosystems and resources creating a barrier to the movement of terrestrial animals and reducing sizes of some ecosystems and resources. Participants in the meetings raised concerns that "*the infrastructure [had] also been seen to affect wildlife movements, for example with animals congregating along the highway*". To mitigate this the SGR contractors provided underpasses and bridges to allow wildlife to move freely. However, local communities have settled within these underpasses and under bridges. During this study, illegal settlements were observed along the SGR section bisecting Tsavo East and West National parks, further blocking wildlife movement across the SGR. This was further reinforced by meeting participants who observed that "a *lot of underpasses have been blocked by the proliferation of illegal settlements and the conversion of land to agriculture*". Due to the sustained construction activities along the SGR and its use, most animals, especially elephants were observed to have changed their behaviour and responses as observed by one local leader that "*some animals such as elephants have become more aggressive as they interpret traffic noise as an indicator of the presence of humans and consequently appear to be on the defensive*". One of the consequences of the barrier effect is the emergence and intensification of human-wildlife conflicts (HWC) as animals move away from railways and roads to surrounding communities, thereby reducing the buffer between wildlife areas and human communities. Supporting this observation, some respondents confirmed that "*there had been an observed increase in human-wildlife conflict which may not necessarily be attributed to the SGR*".

## Ecosystem destruction

Destruction of the ecosystem due to the SGR and related activities was the least dominant issue of concern to our participants. The construction of the SGR together with the associated quarrying activities resulted in the removal of forest or vegetation cover, destruction of water sources such as rivers, wetlands, grasslands, and parts of protected areas Participants observed that activities along the SGR "*modified or disturbed the natural ecosystems especially low-lying, poorly drained land in areas around Kibwezi, Mombasa ad Voi*". Of concern was the conversion of land [agricultural and grazing] to settlement and real estates. During a meeting in Kibwezi, the participants stated that they "*had observed a number of changes in land use in the area, mainly sub-division of land due to population pressure and development of the area through the railway and road network*". Whereas participants in Narok noted that "*real estate development was picking up in Suswa. In addition, land sales were high especially around the proposed [SGR] station in Suswa, mainly in anticipation of the [Naivasha] dry port*". Our team observed that wetlands around Kitengela and Kiboko were blocked off and damaged, thereby affecting natural water flow (Fig 3B). Similarly, we observed that forests in Kibwezi at the KEFRI station had been cleared to create way for the construction of the railway.

**Ecosystem degradation, fragmentation and destruction.** Although the actual land area directly affected by the SGR may only cover a small area along the corridor, a much wider area supports a wide range of ecologically fragile and important ecosystems in Kenya. The SGR has clearly had some direct ecological impacts emanating from its construction and subsequent operation. In particular, the SGR has contributed to wider ecosystem degradation and fragmentation. Although similar impacts have been reported by road and railway ecologists through quantitative scientific approaches in different contexts, this study demonstrates that

these impacts can be understood by engaging with a range of different stakeholders from diverse backgrounds, that bring together a range of experiences and expertise, even when and where the infrastructure development is relatively new, as is the case in Kenya.

The majority of participants in this study identified ecosystem degradation as the main impact around the SGR. Within the context of transportation infrastructure, degradation may arise from disturbance, pollution and contamination of soil, water and air and disruption of natural processes [20]. The SGR construction was accompanied by activities such as soil compaction, excavation and movement of soil from one location to another to erect the embankments. These activities altered and created barriers to natural processes including natural hydrology and animal migration routes. Previous research demonstrates how wildlife residing close to the infrastructure is affected by the traffic noise, vibrations, chemical pollution and human presence [18, 22]. Although infrastructure engineers rely on technical considerations when designing and implementing these projects, local knowledge and experiences of critical ecosystems can contribute to the identification and protection of these ecosystems. In areas where engineers and scientists have no prior knowledge and experience of potential impacts, stakeholder perceptions can play an important role in providing baseline information to design, implement and monitor impact mitigation activities.

Although little scientific research has been conducted on the changing nature of wildlife behaviour and railways, especially in Africa, we found that elephants had displayed early signs of behavioural modification in response to the activities around and within the SGR corridor particularly around Tsavo in Voi. These are consistent with behavioural adaptations observed among other species including shifting their home ranges or altering their movement patterns away from areas with high infrastructure densities (e.g. [42–44]). Barnes *et al.* [45] reported that elephants (*Loxodonta africana*) in north-eastern Gabon preferred sites away from both roads and villages. However, this does not hold true for all species, for example, Coleman and Fraser [46] reported that black vultures (*Coragyps atratus*) and turkey vultures (*Cathartes aura*) preferred home ranges in areas with greater road densities perhaps due to availability of food. The ability of stakeholders to articulate these provides a key signal towards systematic monitoring of such changes by railway ecologists in Kenya.

The SGR, by its linear form, cuts across watersheds and drainage basins thereby altering and modifying the local natural hydrological environment. The SGR contractors rerouted or concentrated surface runoff to the underpasses, inevitably increasing the volumes and speed of the flow. This has resulted in flooding, soil erosion, channel modification, and siltation of streams. A significant environmental impact in the SGR is soil erosion, sedimentation and resultant flooding. Indeed, clearing vegetation along and adjacent to the infrastructure can lead to soil erosion and sediment inputs to watercourses [47]. These degraded ecosystems might depict altered microclimatic conditions with risks of flooding and landslides [2]. Meanwhile, underground water sources can be degraded by runoff and hazardous material spills that contaminate aquifers [25]. Furthermore, our study raises concern about the use of embankments to raise the SGR as this inevitably results in rerouting of surface runoff to avoid cutting through the earth embankments. These concerns challenge the conventional infrastructural engineering designs that rely on data and experiences from other areas. The SGR design, alignment and routing was based on existing designs and expertise from China. This approach can draw parallels with challenges around technological transfers that do not take into account local ecological and social context. Engaging with local communities and stakeholders can offer insights into local and ecologically sensitive designs with minimal impacts.

The introduction and spread of invasive plant species due to the recent large scale expansion of the transportation infrastructure have become a major global concern [48–50]. As suggested by Hulme [27] and Catford *et al.* [26], railway infrastructure can act as a corridor for

the natural dispersal of non-native plant species as was observed in different locations along the SGR corridor. Studies across Africa, US and Europe have shown that railway verges and embankments host a high diversity of non-native species [51–53]. In most cases, railway verges are regularly mowed or cleared resulting in open spaces suitable for invasive species [51, 54]. This creation of new habitats opens up new niches that can be exploited by invasive plant species that can migrate quickly along the linear infrastructure. Early detection through consultations with local communities and stakeholders can offer opportunities for effective management and eradication.

Some of the railway impacts required long-term monitoring to establish discernible patterns such as physical disturbance and disruption of the continuous vegetative community, structure and function of ecosystems, and movement, migration and survival of resident wildlife species [13, 14, 17]. Such disruptions may result in isolation of populations, gene flow restrictions and loss of biodiversity [13, 17, 22, 55, 56]. Furthermore, studies elsewhere have reported mortality of a variety of mammal species such as grizzly bears (*Ursus arctos*) [57], moose (*Alces alces*) [58], elephants (*Loxodonta spp*) [59–63] and frogs (*Anura spp*) [64] for decades. These studies have involved long-term systematic monitoring and reporting of the impacts. Given the short span between construction and operation of the SGR, it is not surprising that participants in this study only reported fewer cases and had no clear appreciation of the long term impacts. However, stakeholder input offered a clear need for more systematic monitoring of these impacts to ascertain their occurrence and severity in Kenya.

The reduction or destruction of ecosystems and replacement with non-natural habitats is a key component of infrastructure development [20]. Although the SGR contractors undertook post-construction rehabilitation of the degraded environment, the original natural characteristics of the land were eliminated, replaced or modified, and there is a significant impact on the landscape of railway infrastructure [21, 29]. Communities relying on wetlands and rivers in Voi, Kibwezi, Tuala and Narok areas lost access to these critical resources, and it is not yet clear how other ecosystem services could be lost in the longer term. Clearly the infrastructure impacts on ecosystems will impact on the value and delivery of natural capital, particularly those associated with water and wildlife impacts.

The discussions above point to the existence of negative impacts of the SGR to the ecological systems along the corridor. These impacts would normally be identified and effective mitigation measures put in place during and after the construction of the SGR through Environmental and Social Impact Assessment (ESIA). During this study, we confirmed that ESIA for the two phases of the SGR were conducted and final reports written to facilitate licencing by the National Environment Management Authority (NEMA), the government regulator [36, 37]. Kenya's ESIA framework is fairly comprehensive and is anchored on sound legislative and institutional set-up, designed to protect both the social and ecological systems and emphasizes public participation in the ESIA process [65]. However, the persistence and emergence of potential ecological impacts coupled with the likely ineffectiveness of mitigation measures outlined by the participants in this study, point to challenges with public participation and little oversight both in the ESIA process and implementation of the development projects.

## Conclusion

The construction of the SGR has led to major impacts on ecosystem, particularly degradation fragmentation and to a less extent ecosystem destruction. Landscape modification by the SGR construction has resulted in increased soil erosion, land degradation, flooding, sedimentation of water bodies, habitat destruction and impeding wildlife movements. It is therefore

recommended that linear infrastructure projects, like the SGR, develop sustainable and ecologically sensitive measures to mitigate against these impacts. For example, underpasses at the right density and of the right size will maintain wildlife movements, water courses can be channelled and redirected, invasive plants may need to have direct control measures implemented to try and stop their spread. Furthermore, a wholesome assessment of the environmental impacts of transportation infrastructure which involves extensive engagement of stakeholders is key for designing and implementing inclusive, resilient and sustainable infrastructure so that the development gains are maximised while the ecosystem impacts are minimised. Further research is needed to quantify the ecological impacts of the SGR, and other transport infrastructure on ecosystems and the associated natural capital with a view of establishing mitigation measures that would promote more sustainable and resilient developments in future.

## Supporting information

**S1 Appendix.**
(DOCX)

**S2 Appendix.**
(DOCX)

## Acknowledgments

This paper is part of a larger study under the Development Corridors Partnership (DCP) project. The DCP aims to generate decision-relevant evidence and feeding into key decision-making processes in order to improve the sustainable development outcomes and investments in infrastructure projects. The researchers would like to thank all the stakeholders who contributed to the study.

## Author Contributions

**Conceptualization:** Tobias Ochieng Nyumba, Catherine Chebet Sang, Daniel Ochieng Olago, Robert Marchant, Lucy Waruingi, Francis Kago, George Owira, Rosemary Barasa, Sherlyne Omangi.

**Data curation:** Tobias Ochieng Nyumba, Catherine Chebet Sang, Yvonne Githiora, Francis Kago, Mary Mwangi, George Owira, Rosemary Barasa, Sherlyne Omangi.

**Formal analysis:** Tobias Ochieng Nyumba.

**Funding acquisition:** Daniel Ochieng Olago, Robert Marchant, Lucy Waruingi.

**Investigation:** Tobias Ochieng Nyumba, Catherine Chebet Sang, Daniel Ochieng Olago, Lucy Waruingi, Yvonne Githiora, Francis Kago, Mary Mwangi, Rosemary Barasa, Sherlyne Omangi.

**Methodology:** Tobias Ochieng Nyumba, Catherine Chebet Sang, Daniel Ochieng Olago, Robert Marchant, Lucy Waruingi, Yvonne Githiora, Francis Kago, Mary Mwangi, George Owira, Rosemary Barasa, Sherlyne Omangi.

**Project administration:** Tobias Ochieng Nyumba, Catherine Chebet Sang, Daniel Ochieng Olago, Lucy Waruingi, Yvonne Githiora, Francis Kago, Mary Mwangi.

**Resources:** Tobias Ochieng Nyumba, Catherine Chebet Sang, Daniel Ochieng Olago, Lucy Waruingi.

**Software:** Tobias Ochieng Nyumba.

**Supervision:** Tobias Ochieng Nyumba, Catherine Chebet Sang, Daniel Ochieng Olago, Robert Marchant, Lucy Waruingi.

**Validation:** Tobias Ochieng Nyumba, Catherine Chebet Sang, Daniel Ochieng Olago, Robert Marchant, Lucy Waruingi.

**Visualization:** Tobias Ochieng Nyumba, Catherine Chebet Sang.

**Writing – original draft:** Tobias Ochieng Nyumba, Catherine Chebet Sang.

**Writing – review & editing:** Tobias Ochieng Nyumba, Catherine Chebet Sang, Daniel Ochieng Olago, Robert Marchant, Lucy Waruingi.

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
