## [Decision Letter · Decision Letter 0]

19 Jan 2021

Assessing the ecological impacts of transportation infrastructure development: a reconnaissance study of the Standard Gauge Railway in Kenya

PONE-D-20-29620

Dear Dr. Nyumba,

We’re pleased to inform you that your manuscript has been judged scientifically suitable for publication and will be formally accepted for publication once it meets all outstanding technical requirements.

Kind regards,

Stefan Lötters

Academic Editor

PLOS ONE

Reviewers' comments:

Reviewer's Responses to Questions

**Comments to the Author**

1. Is the manuscript technically sound, and do the data support the conclusions?

Reviewer #1: Yes

2. Has the statistical analysis been performed appropriately and rigorously? 

Reviewer #1: Yes

3. Have the authors made all data underlying the findings in their manuscript fully available?

Reviewer #1: Yes

4. Is the manuscript presented in an intelligible fashion and written in standard English?

Reviewer #1: Yes

5. Review Comments to the Author

Reviewer #1: And important topic and interesting approach. Not only is more community consultation necessary, but also more education and knowledge sharing for local communities is essential. There should be a closer look at local knowledge and ecological impacts when solving ecological problems as impactful as linear infrastructure projects.

6. PLOS authors have the option to publish the peer review history of their article (what does this mean?). If published, this will include your full peer review and any attached files.

Reviewer #1: **Yes: **Melly Reuling

---

## [Editor Report · Acceptance letter]

21 Jan 2021

PONE-D-20-29620 

Assessing the ecological impacts of transportation infrastructure development: a reconnaissance study of the Standard Gauge Railway in Kenya 

Dear Dr. Nyumba:

I'm pleased to inform you that your manuscript has been deemed suitable for publication in PLOS ONE. Congratulations! Your manuscript is now with our production department. 

Kind regards, 

on behalf of

Prof. Dr. Stefan Lötters 

Academic Editor

PLOS ONE